# Food Enrichment with *Glycyrrhiza glabra* Extract Suppresses ACE2 mRNA and Protein Expression in Rats—Possible Implications for COVID-19

**DOI:** 10.3390/nu13072321

**Published:** 2021-07-06

**Authors:** Daniela Jezova, Peter Karailiev, Lucia Karailievova, Agnesa Puhova, Harald Murck

**Affiliations:** 1Institute of Experimental Endocrinology, Biomedical Research Center, Slovak Academy of Sciences, 84505 Bratislava, Slovakia; Daniela.Jezova@savba.sk (D.J.); peter.karailiev@savba.sk (P.K.); lucia.karailievova@savba.sk (L.K.); Agnesa.Puhova@savba.sk (A.P.); 2Department of Psychiatry and Psychotherapy, Philipps-University Marburg, 35039 Marburg, Germany; 3Murck-Neuroscience, Westfield, NJ 07090, USA

**Keywords:** COVID-19, glycyrrhizin, mineralocorticoid receptor, toll like receptor 4, angiotensin converting enzyme, aldosterone

## Abstract

Angiotensin converting enzyme 2 (ACE2) is a key entry point of severe acute respiratory syndrome coronavirus 2 (SARS-CoV-2) virus known to induce Coronavirus disease 2019 (COVID-19). We have recently outlined a concept to reduce ACE2 expression by the administration of glycyrrhizin, a component of *Glycyrrhiza glabra* extract, via its inhibitory activity on 11beta hydroxysteroid dehydrogenase type 2 (11betaHSD2) and resulting activation of mineralocorticoid receptor (MR). We hypothesized that in organs such as the ileum, which co-express 11betaHSD2, MR and ACE2, the expression of ACE2 would be suppressed. We studied organ tissues from an experiment originally designed to address the effects of *Glycyrrhiza glabra* extract on stress response. Male Sprague Dawley rats were left undisturbed or exposed to chronic mild stress for five weeks. For the last two weeks, animals continued with a placebo diet or received a diet containing extract of *Glycyrrhiza glabra* root at a dose of 150 mg/kg of body weight/day. Quantitative PCR measurements showed a significant decrease in gene expression of ACE2 in the small intestine of rats fed with diet containing *Glycyrrhiza glabra* extract. This effect was independent of the stress condition and failed to be observed in non-target tissues, namely the heart and the brain cortex. In the small intestine we also confirmed the reduction of ACE2 at the protein level. Present findings provide evidence to support the hypothesis that *Glycyrrhiza glabra* extract may reduce an entry point of SARS-CoV-2. Whether this phenomenon, when confirmed in additional studies, is linked to the susceptibility of cells to the virus requires further studies.

## 1. Introduction

The coronavirus pandemic 2019 (COVID-19) has clearly revealed the need to search for new therapeutic options including natural products as food supplements [1,2]. The angiotensin converting enzyme 2 (ACE2) serves as an entry point for the severe acute respiratory syndrome coronavirus 2 (SARS-CoV-2), which leads to COVID-19. Therefore, reducing ACE2 expression would reduce the number of access points of the virus to the body during primary infection, and potentially the spread inside the body. Cells which are susceptible to infection with SARS-CoV-2 appear to be primarily type II pneumocytes, intestinal absorptive enterocytes, and nasal goblet secretory cells [3]. The identification of mechanisms to reduce membrane ACE2 expression at these cells may be valuable.

We have recently proposed that glycyrrhizin, a key component of an extract from *Glycyrrhiza glabra*, may have such an effect [4]. A beneficial effect and potential mechanisms of action of glycyrrhizin, or components from *Glycyrrhiza glabra,* have been reviewed by several groups independently of our original suggestion [5,6,7]. Glycyrrhizin is metabolized into the systemically active metabolite glycyrrhetinic acid. Via this metabolite, glycyrrhizin inhibits an enzyme called 11-beta-hydroxysteroid dehydrogenase type 2 (11betaHSD2) [8]. Its inhibition allows cortisol to access mineralocorticoid receptors (MR) in aldosterone specific peripheral tissues, including the kidney, lung, intestinal, nasal and endothelial cells, in which it would otherwise have been prevented from doing so. In other words, an inhibition of this enzyme leads to an aldosterone-like activation at the MR by cortisol and may resemble the effects of high aldosterone levels in these organs. This may be relevant, as compounds which reduce plasma aldosterone, including angiotensin-converting enzyme inhibitors and angiotensin receptor antagonists, increase the expression of ACE2 [9]. Conversely, MR activation leads to a downregulation of ACE2, as demonstrated in the kidney [10]. 

Such action could therefore be a mechanism which could be employed to reduce ACE2 expression and, therefore, access of the virus to specific cells. ACE2 is an enzyme [3,11,12], not a receptor, but serves as a receptor for viral particles. This is important to keep in mind in order to avoid confusion regarding nomenclature. The confusion between the term “ACE2” and “ACE2 receptor” present in the literature arose due to the fact that ACE2 serves as a receptor for SARS-CoV-2, therefore being correctly called the “SARS-CoV-2 receptor” and not the “ACE2 receptor” [3,11,12]. A relevant tissue expresses, besides ACE2, both 11betaHSD2 and the MR. This includes lung and nasal, as well as intestinal epithelial cells. From a mechanistic perspective it is important that the small intestine, in particular ileum cells, co-expresses MR, 11betaHSD2 and ACE2, implying that the ileum could serve as an entry point for Cov-SARS-2 and be a target for 11betaHSD2 inhibition; it is at least a model organ to test for the effect of glycyrrhizin on ACE2. 

The downstream consequences of reduced ACE2 expression are somewhat controversial. ACE2 activity is generally protective, including for lung tissue [13]. It protects by converting angiotensin II to angiotensin1–7 [14] as well as by suppressing the consequences of the activation of the receptor for endotoxin (LPS), i.e., the toll-like receptor 4 (TLR4) and, as a consequence, related inflammation in the lung (endotoxin storm) [15]—ACE2 overexpression inhibited the LPS induced inflammation in the mentioned study. Therefore, the reduced expression of ACE2 could be regarded as concerning. The anti-inflammatory ACE2-system is, however, balanced against the pro-inflammatory classical ACE [16], which leads to an increase in the pro-inflammatory mediator angiotensin II. Inhibition of 11betaHSD2 by glycyrrhizin or glycyrrhetinic acid suppresses the classical renin–angiotensin–aldosterone system (RAAS), i.e., reduces the plasma concentrations of renin, angiotensin and aldosterone [8,17] and increases cortisol/corticosterone locally [18]. This inhibition of the classical RAAS and activation of glucocorticoid receptors may therefore add to a potential beneficial effect of glycyrrhizin via the reduction of the pro-inflammatory angiotensin II [19]. Furthermore, a direct anti-inflammatory effect of *Glycyrrhiza glabra* extract and glycyrrhizin, via inhibition of TLR4 and inhibition of the release of high mobility group box 1 (HMGB1), has been described [20,21]. Such actions would counteract the consequences of ACE2 suppression on inflammation. In accordance, glycyrrhizin has protective effects in acute respiratory distress syndrome induced by the TLR4 activator LPS in mice [20]. 

The objective of this retrospective analysis is to explore the capability of *Glycyrrhiza glabra* extract to reduce ACE2 expression in the small intestine (ileum), as a target tissue with active 11betaHSD2 and MR expression, in comparison to non-target tissues (brain and heart), to provide mechanistical evidence that *Glycyrrhiza glabra* extract may have clinical benefits via reduced expression of ACE2. The tissues were obtained in an already performed study in rats, which was designed to identify the effects of *Glycyrrhiza glabra* extract on the stress response.

## 2. Materials and Methods

### 2.1. Animals

Forty-eight male Sprague-Dawley rats (Velaz, Prague, Czech Republic) weighing 225–250 g at the beginning of the experiments were used. The rats were allowed to habituate to the housing facility for 5 days. The animals were housed under standard laboratory conditions with free access to food and water. A constant 12:12 h light–dark cycle was maintained with light on at 07.00 h and off at 19.00 h. Temperature was maintained at 22 ± 2 °C and humidity at 55 ± 10%. All experimental procedures were approved by the Animal Health and Animal Welfare Division of the State Veterinary and Food Administration of the Slovak Republic (permission No. Ro 2291/18-221/3) and conformed to the NIH Guidelines for Care and Use of Laboratory Animals.

### 2.2. Study Design

This was not originally designed to study the effects of *Glycyrrhiza glabra* root extract on ACE2 expression, but instead, was based on data from an already performed study in rats, which addressed the effects of *Glycyrrhiza glabra* extract on the stress response (report in preparation). However, given the urgent need to identify treatments for COVID-19, we were motivated to address a hypothesis, which was formulated earlier [4]. With respect to the nature of this study, obvious limitations had to be accepted, in particular the unavailability of lung tissue at the time of raising these questions. 

Following the habituation to the animal facility, the rats were randomly assigned to the control groups (*n* = 24) and to groups of animals exposed to chronic mild stress (*n* = 24). The model of chronic mild stress was based on seven different stress stimuli [22]. These involved social isolation (animal alone in the cage), unknown cage mate (the animal shared the cage with a rat from another cage), stroboscopic light (light flashes with frequency of 5 flashes/s), cage tilt (cages were tilted to 45 degrees from the horizontal), wet cage (water surface reached 2 cm above the bottom of the cage), continuous lighting (lighting for 24 h) and water deprivation. These stimuli were applied for 12 h each, in a randomized order, i.e., two conditions per day for 5 weeks. Control animals were housed undisturbed in a different room under the same light and temperature conditions. They had free access to food and water. 

### 2.3. Treatment

The control rats as well as rats exposed to chronic mild stress were randomly assigned to one of the two groups: animals fed a diet with extract of *Glycyrrhiza glabra* (*n* = 12) and animals fed a placebo diet (*n* = 12). The extract of *Glycyrrhiza glabra* roots (Gall-Pharma GmbH, Judenburg, Austria) (Batch. no. P17092209) contained 6.25% of glycyrrhizinic acid. Water was used as a solvent during the extraction. The extract was mixed into the placebo diet at a dose of 150 mg/kg/day (SSNIFF Specialdiäten GmbH, Soest, Germany). The dose was selected according to [23]. In this study behavioral effects of *Glycyrrhiza glabra* extract were observed, ensuring that a relevant plasma level had been reached. Unfortunately, we were not able to measure plasma levels directly. The placebo diet (SSNIFF Specialdiäten GmbH, Soest, Germany) consisted of carbohydrates (65%), protein (24%) and fat (11%).

As mentioned above, the experiments lasted for 5 weeks. All animals received normal control diet for the first 3 weeks. The rats assigned for *Glycyrrhiza glabra* were fed the diet containing extract of *Glycyrrhiza glabra* for the next two weeks. 

### 2.4. Organ Collection

Following 5 weeks of experimental procedures, the animals were quickly decapitated with a guillotine between 08.00 and 10.30 h in the morning. The brain was quickly removed from the skull and the prefrontal cortex was dissected on an ice-cold plate. The heart was removed and rinsed in 0.9% NaCl solution. The left heart ventricle was cut from the whole heart. Subsequently, the small intestine was removed from the body and all samples were frozen and stored at −70 °C until analyzed. 

### 2.5. ACE2 mRNA and Protein Quantification

The gene expression of ACE2 was measured in the small intestine, the prefrontal cortex and the left heart ventricle by quantitative PCR. In the case of small intestine, its content was removed before tissue homogenization. Total RNA extraction, transcription of mRNA into cDNA as well as gene expression quantification was performed as described previously [24]. Primer BLAST NCBI software was used to design primers specific for the studied genes as well as reference genes (Table 1).

The concentration of ACE2 protein in the small intestine was determined by Rat Ace2 ELISA Kit (cat. no. ER0609, FineTest, Wuhan Fine Biotech Co., Ltd., Wuhan, Hubei, China). The intestines were thawed and their contents were removed. The samples were then frozen in liquid nitrogen in a mortar and pulverized by a pestle. A pre-test with the mentioned ELISA kit was performed to determine the optimal amount of tissue to be used in the subsequent analysis. We found out that 12.5 mg of powdered tissue (1/8 of the recommended amount) was the most favourable amount that fitted well into the kit’s standard curve. The powdered tissue was suspended in 900 µL of PBS (according to the manufacturer’s protocol) and was left at room temperature for 30 min. The samples were centrifuged at 5000× *g* for 5 min at 4 °C. From this point, the analysis was performed according to the manufacturer’s protocol with 100 µL of sample supernatant put into the plate wells. The results are expressed as ng/mg of tissue.

### 2.6. Hormone Measurements

The trunk blood was collected and the plasma used for the analyses. Plasma corticosterone was measured by double-antibody radioimmunoassay (MP Biomedicals, Solon, OH, USA). Both intra- and inter-assay coefficients of variation (CVs) were <5%. Plasma renin activity was measured using angiotensin I radioimmunoassay kit (Immunotech, Marseille, France). The intra- and inter-assay CVs were 11.3% and 20.9%, respectively. Serum aldosterone was analyzed by a coated-tube radioimmunoassay (RIAZENco Aldosterone kit, ZenTech, Liège, Belgium), according to the manufacturer’s instructions. The intra- and inter-assay CVs were 3.8% and 6.2%, respectively. 

### 2.7. Statistical Analyses

The software package used for the statistical analysis was Statistica 7 (Statsoft, Tulsa, OK, USA). The values were checked for normality of distribution using the Shapiro-Wilks test. Data not normally distributed were Winsorized to normalize the distributions before analyses. Data from the gene and protein expression of ACE2 were analyzed by two way analysis of variance (ANOVA) with main factors of treatment (*Glycyrrhiza glabra* extract vs. placebo) and stress (chronic mild stress vs. control). Only the data of food intake, which were recorded in time, were analysed by a repeated measures ANOVA for factor time, treatment and stress. For post hoc comparisons, the Tukey post hoc test was chosen as this test is appropriate for two-way ANOVA and is stricter in comparison with other tests, such as Fisher least significant difference (LSD). Results are expressed as means ± standard error of the mean (SEM). The overall level of statistical significance was set as *p* < 0.05. 

## 3. Results

In the small intestine, a tissue with known high activity of 11β-HSD2, the gene expression of ACE2 was significantly lower in rats fed the diet with *Glycyrrhiza glabra* extract compared to rats fed the placebo diet (Figure 1A). Two-way ANOVA revealed a significant main effect of treatment (F(_1,44_) = 4.41; *p* = 0.0415) on concentrations of mRNA coding for ACE2 in the small intestine. The effect of stress was not statistically significant. 

To verify if the observed changes in intestinal ACE2 mRNA production correlate with the protein levels, ACE2 protein concentrations were measured by ELISA. The protein concentrations of ACE2 in the small intestine were significantly lower in rats fed the diet with *Glycyrrhiza glabra* extract compared to rats fed the placebo diet (Figure 1B). Two way ANOVA revealed a significant main effect of treatment (F(_1,44_) = 4.46, *p* = 0.0403) on concentrations of ACE2 protein in the small intestine. The effect of stress was not statistically significant.

Concentrations of mRNA coding for ACE2 in the left heart ventricle were not affected by *Glycyrrhiza glabra* extract treatment (Figure 1C). The difference between the concentrations in the stressed groups was not statistically significant. Similarly, the gene expression of ACE2 in the prefrontal cortex was unchanged (Figure 1D).

The statistical analysis of corticosterone concentrations in plasma revealed a significant interaction between the main factors of treatment and stress (F(_1,44_) = 10.23; *p* = 0.0030). The post hoc analysis showed that plasma corticosterone was significantly increased in the control group, which received *Glycyrrhiza glabra* extract vs. placebo (*p* = 0.0186; Figure 2a). This was not found in the stressed group, which showed a numerical reduction of corticosterone with the administration of *Glycyrrhiza glabra* extract which failed to be statistically significant. Aldosterone concentrations were numerically but not significantly reduced (Figure 2b). Plasma renin activity was unchanged (Figure 2c).

To check the potential influence of *Glycyrrhiza glabra* extract on food intake and thus on the dose of the drug ingested, we measured the food intake in two 2-day time intervals. The food intake was not significantly affected by the treatment. Chronic mild stress induced a significant reduction of food intake. Repeated measures ANOVA revealed a significant main effect of stress (F(_1,44_) = 43.43; *p* < 0.001), as well as time (F(_1,44_) = 40.59; *p* < 0.001) on food intake (Table 2). 

## 4. Discussion

The main finding of this study is the support for the hypothesis [4] that the treatment with *Glycyrrhiza glabra* reduces the expression of both gene and protein of ACE2 in tissue, which co-expresses 11betaHSD2 and MR, and may therefore reduce the cellular uptake and spread of SARS-CoV-2. Mechanistically, corticosterone acts as a mineralocorticoid to activate the MR in this situation and as a consequence reduces ACE2 expression. This effect was independent of the stress condition and failed to be observed in non-target tissues, such as the heart and the brain. Observed increase in plasma corticosterone in the control condition with the treatment of *Glycyrrhiza glabra* extract confirms target engagement. 

The observed increase in glucocorticoid concentrations may also be partially responsible for an expected clinical benefit, as the synthetic glucocorticoid dexamethasone is clinically effective against COVID-19 symptoms [25], which is considered to be mediated via its anti-inflammatory effect. An alternative pathway of glycyrrhizin to affect inflammatory processes is via its activity to modify gut microbiota [26]. It has been reported that gut bacteria metabolize steroids into compounds, which modifies 11betaHSD2 [27] and may therefore have an impact on local ACE2 expression. Interestingly, ACE2 expression appears to affect the gut microbiome and, in turn, changes in gut microbiota may lead to changes in ACE2 expression [28]. 

The reduction in ACE2 mRNA levels as well as the ACE2 protein content in the small intestine revealed by feeding the rats with *Glycyrrhiza glabra* extract supplemented diet in the present study represents a novel original finding. Consistently with our hypothesis, the expression of ACE2 in tissues without evident 11betaHSD2 activity, such as the heart and brain cortex [29,30], remained unchanged. There are several review articles suggesting potential positive action of natural products on both prevention and treatment of the disease induced by SARS-CoV-2 [4,31,32,33,34,35,36]. With respect to supporting experimental evidence, effects of *Glycyrrhiza glabra* root extract on ACE2 expression have not been reported so far but the present data are consistent with the action of an extract of another plant, namely *Glycyrrhiza uralensis,* in the lung tissue of mice [37]. Other supporting data show that extract of *Glycyrrhiza glabra* or its main components may affect affinity interactions with ACE2 and/or viral proteases [38,39,40]. 

It may by suggested that reduced ACE2 expression induced by treatment with *Glycyrrhiza glabra* extract could have several beneficial implications for future clinical research. The most important future direction is to verify the protective effect of the extract and/or its main component glycyrrhizin against the entry of SARS-CoV-2 into the cells. Indeed, our preliminary results have shown a direct antiviral effect of glycyrrhizin on the replication of isolated human SARS-CoV-2 in a Vero E6 cell culture in a plaque-reduction inhibition test. These experiments revealed that depending on the concentration of glycyrrhizin added to the cell culture media, an inhibition of SARS-CoV-2 replication down to the detection limit of the assay was observed [41]. Supporting results were also reported by others [42]. Glycyrrhizin exerted a stronger effect when it was present in the cell culture media during the infection and subsequent incubation than when it was added after the virus infection step. Thus, the whole extract from *Glycyrrhiza glabra* or glycyrrhizin, which are generally regarded as safe, are promising dietary ingredients to help with prevention or early treatment of COVID-19 [41]. Preliminary clinical data on the positive effects of glycyrrhizin [43] or glycyrrhizin containing extracts [44] to treat patients with COVID-19 support the mechanistic data outlined here. 

To discuss the present findings in the context of pulmonary diseases, the primarily targeted cells for glycyrrhizin treatment are lung epithelial cells (type II pneumocytes), which express 11betaHSD, both type 1 and 2 [45,46]. However, 11betaHSD2 appears to be upregulated in acute respiratory distress syndrome [47]. Aldosterone leads to an increase in alveolar clearance via an interaction with MR [48]. These observations support a role of 11betaHSD2 inhibition and resultant MR activation in lung protection from inflammatory stimuli. An independent confirmation of the role of MR in lung protection comes from clinical observations that the MR agonist fludrocortisone in combination with corticosteroids leads to a better clinical outcome in septic shock than corticosteroids alone [49]. It should, however, be noted that aldosterone via MR activation has, in many situations, pro-inflammatory effects, depending on tissue and other factors. 

The present results are important also with respect to gastrointestinal problems related to COVID-19. It has been reported that a significant percentage of patients with COVID-19 experience gastrointestinal symptoms [50]. Approximately half of patients with confirmed COVID-19 have shown measurable SARS-CoV-2 RNA in their stool samples [51]. It is known that intestinal tissues contain the coronaviruses for weeks after the initial upper respiratory syndrome. Indeed, viral nucleic acid was found to be present in the feces after pharyngeal swabs became negative [52]. The present findings of reduced intestinal ACE2 expression by dietary supplementation with *Glycyrrhiza glabra* extract might attenuate virus accumulation in the gastrointestinal tract and thus contribute to the prevention of potential fecal–oral transmission of SARS-CoV-2.

An obvious limitation of this study is the lack of lung tissue as the original experimental design had a different goal. Another limitation of the presented concept is its focus on ACE2 only as an entry point for SARS-CoV-2. It has to be noted that other proteins, including CD209L and CD147 may serve this purpose. An interaction between these proteins and either the renin–angiotensin–aldosterone system, ACE2 or the effect of glycyrrhizin has not been reported.

## 5. Conclusions

In conclusion, the treatment with *Glycyrrhiza glabra* root extract leads to a significant reduction in the expression of ACE2 in the small intestine, which may serve as an entry point of SARS CoV-2. An important aspect of the current study is to motivate additional work, which needs to be performed to provide more conclusive evidence. Whether a similar effect exists in the lungs needs to be further explored, but it is plausible, given a similar receptor constellation of ACE2, 11betaHSD2 and MR. Whether this phenomenon, when confirmed in additional studies, is linked to the susceptibility of cells to the virus requires further studies.

## Figures and Tables

**Figure 1 nutrients-13-02321-f001:**
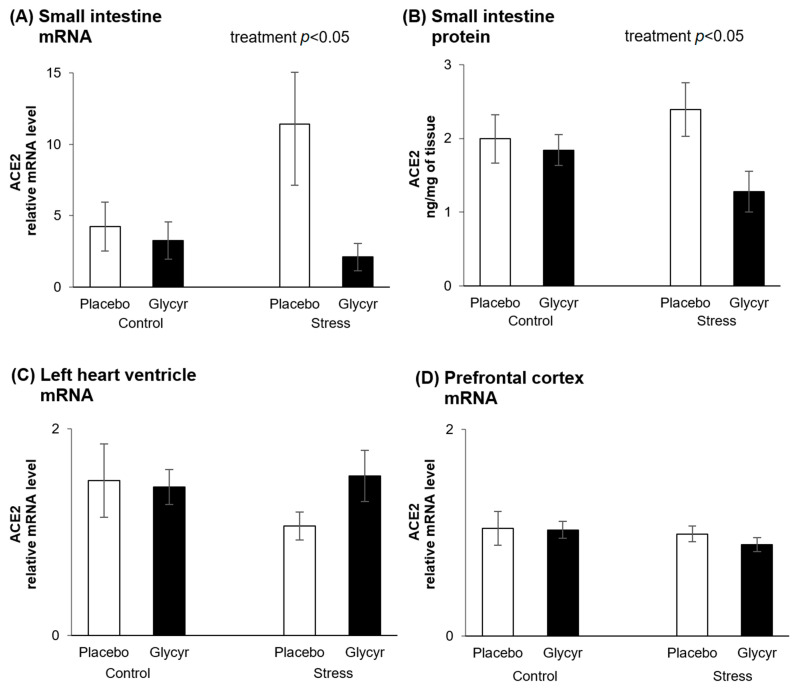
ACE2 gene expression in the small intestine (**A**), ACE2 protein expression in the small intestine (**B**), ACE2 gene expression in the left heart ventricle (**C**) and ACE2 gene expression in the prefrontal cortex (**D**) of rats treated with *Glycyrrhiza glabra* extract or placebo with or without exposure to chronic mild stress. Each value represents mean ± standard error of the mean (SEM) (*n* = 12 rats/group). Statistical significance as revealed by two-way ANOVA.

**Figure 2 nutrients-13-02321-f002:**
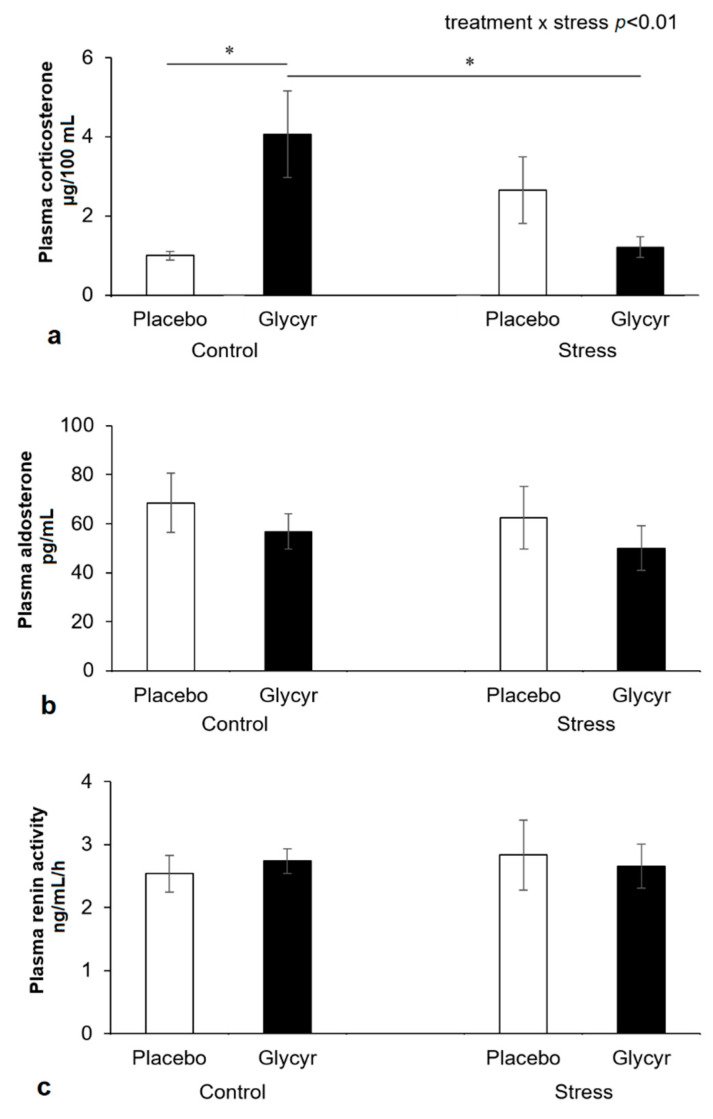
Concentrations of plasma corticosterone (**a**), concentrations of plasma aldosterone (**b**) and plasma renin activity (**c**) in rats treated with *Glycyrrhiza glabra* extract or placebo with or without exposure to chronic mild stress. Each value represents the mean ± standard error of the mean (SEM) (*n* = 12 rats/group). Statistical significance as revealed by two-way ANOVA followed by Tukey post-hoc test: * *p* < 0.05.

**Table 1 nutrients-13-02321-t001:** Oligonucleotide sequences used in quantitative PCR.

Gene	Sense	Sequence 5′→3′
*ACE2*	Forward	ACCCTTCTTACATCAGCCCTACTG
	Reverse	TGTCCAAAACCTACCCCACATAT
*UQCRFS1—reference gene*	Forward	ACAGTGGGCCTGAATGTTCC
	Reverse	CACGGCGATAGTCAGAGAAGTC
*TfR1—reference gene*	Forward	ATACGTTCCCCGTTGTTGAGG
	Reverse	GGCGGAAACTGAGTATGGTTGA
*HPRT1—reference gene*	Forward	CGTCGTGATTAGTGATGATGAAC
	Reverse	CAAGTCTTTCAGTCCTGTCCATAA

*ACE2*: Angiotensin converting enzyme 2; *UQCRFS1*: Ubiquinol-cytochrome c reductase, Rieske iron-sulfur polypeptide 1; *TfR1*: Transferrin receptor protein 1; *HPRT1*: Hypoxanthine Phosphoribosyltransferase 1.

**Table 2 nutrients-13-02321-t002:** The effect of treatment with *Glycyrrhiza glabra* extract on average food intake of stressed and non-stressed animals in selected two-day time intervals. Each value represents the mean ± SEM (*n* = 12 rats/group). Statistical significance as revealed by repeated measures ANOVA.

Food Intake (g)	Group				Statistical Significance
	Control		Stress		
Treatment Day	Placebo	Glycyr	Placebo	Glycyr	
1–2	62.6 ± 1.2	65.2 ± 0.9	56.3 ± 0.6	59.8 ± 1.1	Treatment N.S. Stress *p* < 0.001 time *p* < 0.001
8–9	62.7 ± 0.9	61.7 ± 1.1	57.3 ± 0.8	53.8 ± 1.2	

N.S.: Not significant.

## Data Availability

The data that support the findings of this study are available from the corresponding author upon reasonable request.

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
