# Peer review of "Food Enrichment with Glycyrrhiza glabra Extract Suppresses ACE2 mRNA and Protein Expression in Rats—Possible Implications for COVID-19"

_nutrients, 2021, doi:10.3390/nu13072321_

Round 1

Reviewer 1 Report

The current pandemic responsible for the crippling of the health care system is caused by the novel SARS-CoV-2 in 2019 and leading to coronavirus disease 2019 (COVID-19). The virus enters into humans by attachment of its Spike protein (S) to the ACE receptor present on the lung epithelial cell surface followed by cleavage of S protein by the cellular transmembrane serine protease (TMPRSS2). After entry, the SARS-CoV-2 RNA genome is released into the cytosol, where it highjacks host replication machinery for viral replication, assemblage, as well as the release of new viral particles. The major drug targets that have been identified for SARS-CoV-2 through host-virus interaction studies include 3CLpro, PLpro, RNA-dependent RNA polymerase, and S proteins. Several reports of natural compounds along with synthetic products have displayed promising results and some of them are Tripterygium wilfordii, Pudilan Xiaoyan Oral Liquid, Saponin derivates, Artemisia annua, Glycyrrhiza glabra L., Jinhua Qinggan granules, Xuebijing, and Propolis. Article prepared by Jezova et al., entitled "Food enrichment with Glycyrrhiza glabra extract suppresses ACE2 mRNA and protein expression in rats – possible implications for COVID-19" is a study of the effect of this preparation on a living organism. Authors performed a research in which for the last two weeks, rats were fed with a placebo diet or received a diet containing extract of Glycyrrhiza glabra root at a dose of 150 mg/kg of body weight/day. Quantitative PCR measurements showed a significant decrease in gene expression of ACE2 in the small intestine of rats fed with diet containing Glycyrrhiza glabra extract. This effect was independent of the stress condition and failed to be observed in non-target tissues, namely the heart and the brain cortex. In the small intestine Jezova et al. confirmed the reduction of ACE2 also at the protein level. Present findings provide another evidence to support the hypothesis that Glycyrrhiza glabra extract may reduce an entry point of SARS-CoV-2. While the report is undoubtedly interesting, the authors should rewrite the paper in its entirety, deleting the information that the paper is the FIRST report on the role of the extract in the dietary management of COVID-19. There are over 30 reports on this topic in Pubmed and the Authors should discuss them extensively in the discussion.

Reviewer 2 Report

Comments and Suggestions for Authors:

The purpose of this study is try to explore the capability of Glycyrrhiza glabra extract to reduce ACE2 expression in the small intestine (ileum), as a target tissue with active 11betaHSD2 and MR expression in different tissues. The author pointed out that the treatment results with Glycyrrhiza glabra root extract leads to a significant reduction in the expression of ACE2 in the small intestine, which may serve as an entry point of SARS CoV-2. However, according to the results of the whole paper, there is no clear experimental data that can point out and serve as the entry point for COVID-19, and the author needs to provide further data for supporting evidence.

  1. In the Result part, “Concentrations of mRNA coding for ACE2 in the left heart ventricle were not affected by Glycyrrhiza glabra extract treatment (Fig.1C)”. However, in Fig. 1C, it can be seen that the mRNA expression concentration of ACE2 increased in stress Group treated with Glycyrrhiza Glabra This part is inconsistent with the text description and needs to be clarified again.
  2. In the discussion part, the third paragraph, “Indeed, our preliminary results …glycyrrhizin … SARS-CoV-2 in a Vero E6 cell culture in a plaque-reduction inhibition test” and “Depending on the concentration of glycyrrhizin… an inhibition of SARS-CoV-2 replication … assay was observed”. Whether these contents are proved by experimental data or literature sources?
  3. Please check the authors list on the first page, why there is a “and” at the end?

Reviewer 3 Report

This is an experimentally short paper of certain significance in the context of the SARS-2 pandemics. It adds interesting data to the current efforts in unravelling the mechanisms and significance of ACE2 receptors and potential lead compounds to fight coronavirus.

The introduction clearly sets out the hypothesis and the limitations of the paper, which comes from a different initial hypothesis. It is well written and researched.

Please check the English of the last sentence of the first paragraph. THE identification of the mechanismS...?

Methods.  Are fine except Statistical analysis. Please describe better the way you compare data, particularly elaborate the sentence "repeated measures ANOVA for factor time... looks like not well written plus explain the post ANOVA test chosen (Tukey?) and why was chose over other post ANOVA tests and how it was run (all pairs compared?).

Results.

Page 5 starting sentence change to "To verify IF the observed...production CORRELATES WITH THE PROTEIN LEVELS ".

The last paragraph avoid mentioning ANOVA or Tukey (had to be well explained in the methods section) and talk about the statistical significance or not compared with what...plus when you say "numerical" does it mean that are non-significant differences?

 Page 6 It is not clear what you mean by "Repeated ...revealed a significant MAIN (delete?) effect of stress. The following two sentences just say the same in different words. Please unify these three sentences.

Discussion

Please explain better (not longer) the 'complex' relationship between gut bacteria and G glabra. 

Change italics after G. glabra extract containing diet. Moreover I suggest "G. glabra supplemented diet"

Please, give a valid reference to your preliminary antiviral tests (such as (non published data) or any thesis/dissertation.

page 7 last paragraph change to  "Thus, the aqueous extract  of G. glabra and glycyrrhizin, which are..., are promising dietary INGREDIENTS..."

(Note that a food supplement CANNOT be put forward as prevention or Cure of a disease, only medicinal products can claim this)

Page 8

Delete your first paragraph. It is not necessary for your paper and it may be wrong to establish a relationship between G. glabra (alone) and mood from a combination with antidepressants. 

in the last paragraph (limitations) include the lack of lung tissue.

I encourage the authors to discuss the importance of intestinal tissues in harbouring the coronaviruses for weeks after the initial upper respiratory syndrome, with the virus detected in human stool long after the respiratory symptoms have been resolved. Also how this ingredient may help in diminishing the spread via so-called "fecal-oral transmission.

Round 2

Reviewer 2 Report

Comments and Suggestions for Authors:

Most of the comments have been answered. The authors discussed in more details about the ACE2 expression induced by treatment with Glycyrrhiza glabra extract, which may have several beneficial effects on SARS-CoV-2, and more adequate literatures for reference.

The references format must be consistent. For example, the final reference: Molecules 25(17) correct to Molecules 25(17),3904.
